# Aqueous Humor Analysis in Overlapping Clinical Diagnosis of Cytomegalovirus and Rubella Virus Anterior Uveitis

**DOI:** 10.3390/medicina58081054

**Published:** 2022-08-04

**Authors:** Fabrizio Gozzi, Lucia Belloni, Raffaella Aldigeri, Pietro Gentile, Valentina Mastrofilippo, Luca De Simone, Elena Bolletta, Federica Alessandrello, Martina Bonacini, Stefania Croci, Alessandro Zerbini, Gian Maria Cavallini, Carlo Salvarani, Luca Cimino

**Affiliations:** 1Ocular Immunology Unit, Azienda USL-IRCCS, 42123 Reggio Emilia, Italy; 2Clinical Immunology, Allergy and Advanced Biotechnologies Unit, Azienda USL-IRCCS, 42123 Reggio Emilia, Italy; 3Department of Medicine and Surgery, University of Parma, 43100 Parma, Italy; 4Department of Surgical Sciences, Eye Clinic, University of Cagliari, 09124 Cagliari, Italy; 5Department of Biomedical Sciences, Ophthalmology Clinic, University Hospital of Messina, 98125 Messina, Italy; 6Department of Surgery, Medicine, Dentistry and Morphological Sciences, with Interest in Transplants, Oncology and Regenerative Medicine, University of Modena and Reggio Emilia, 41121 Modena, Italy; 7Rheumatology Unit, Azienda USL-IRCCS of Reggio Emilia, 42122 Reggio Emilia, Italy

**Keywords:** overlapping viral anterior uveitis, cytomegalovirus, rubella virus, Fuchs Uveitis, antibody index

## Abstract

*Background and Objectives:* A cross-sectional single-center study was conducted to investigate the etiology in hypertensive anterior uveitis whose clinical features are not fully distinctive from cytomegalovirus or from rubella virus and to demonstrate the possible coexistence of both these viruses in causing anterior uveitis. *Materials and Methods:* The clinical charts of a cohort of patients with hypertensive viral anterior uveitis of uncertain origin consecutively seen in a single center from 2019 to 2022 were retrospectively reviewed; data on the clinical features, aqueous polymerase chain reaction, and antibody response to cytomegalovirus and rubella virus were collected. *Results:* Forty-three eyes of as many subjects with viral anterior uveitis of uncertain origin were included. Thirty-two patients had an aqueous polymerase chain reaction or antibody index positive to cytomegalovirus only, while 11 cases had an aqueous antibody response to both cytomegalovirus and rubella virus. This latter overlapping group had a statistically significant higher rate of hypochromia and anterior vitritis (*p*-value: 0.02 and < 0.001, respectively). *Conclusions:* The simultaneous presence of intraocular antibodies against cytomegalovirus and rubella virus could redefine the differential diagnosis of hypertensive viral anterior uveitis, demonstrating a possible “converged” immune pathway consisting in a variety of stimuli.

## 1. Introduction

Viral anterior uveitis (VAU) represents a group of uveitis that account for 4.5–18.6% of all uveitis in the Caucasian populations of developed countries [1]. It must be suspected in the presence of granulomatous keratic precipitates (KPs) and elevated intraocular pressure (IOP). The most commonly implicated viruses in VAU include herpes simplex virus (HSV), varicella-zoster virus (VZV), cytomegalovirus (CMV), and rubella virus (RV) [2,3]. Several studies have indicated that each virus has its own predictive features in terms of KPs, endotheliitis, iris atrophy, hypochromia, iris nodules such as Koeppe’s nodules, cataract, and anterior vitritis [4,5,6]. For example, in herpetic anterior uveitis, “mutton fat” KPs in a triangular arrangement (Arlt’s triangle) below the horizontal midline and sectorial iris atrophy are frequent findings, the latter much more extended and defined in the case of VZV [7,8]. However, it is common to find doubtful cases with a similar, mystifying clinical picture—in particular, in the differential diagnoses between CMV and RV anterior uveitis. Therefore, it is often not possible to identify with certainty the viral etiology of the uveitis without resorting to an aqueous humor analysis. Aqueous polymerase chain reaction (PCR) and antibody diagnostics can considerably increase VAU diagnostic sensitivity and specificity [9,10].

RV is currently considered the main causative agent of Fuchs uveitis (FU), first described in the early twentieth century by the homonymous Austrian ophthalmologist [11]. In the early 2000s, Quentin and Reiber first showed that RV-specific antibodies were detected in the anterior chamber in 87% of the eyes affected by FU [12]. Since then, numerous studies have evidenced a tenacious association between RV and FU in predominantly Caucasian populations, thanks to the aqueous/serum ratio quantitative antibody analysis [13,14,15]. RV anterior uveitis is difficult to diagnose by RV RNA detection alone, because positive PCR is not reliable. Indeed, many studies have shown that 10–20% of suspected cases were PCR-positive, whereas 87–100% of AH samples were RV-IgG-positive [12,16,17]. However, while some authors stated that CMV can also cause FU in the Asian population in 16–42% of cases of FU, on closer inspection, the CMV-associated FU cases often present with features that differ from those of RV-associated cases, including different KP morphology or the absence of vitritis [18]. It is important to underline the epidemiology: the prevalence of CMV infection in the Asian population with VAU is higher than that in the West, possibly because of its apparently higher seroprevalence in Asian countries (approximately 69.1–98.6%) than in the West (approximately 41.9–57%) [18,19]. Instead, RV infection is much more diffuse in the Caucasian than in the Asian population. Differing genetic susceptibilities or pathogenic strains of these viruses may give rise to this geographic disparity [19]. In particular, different ethnic groups may imply the presence of a distinct and specific cytokine profile implicated in the pathogenesis of FU; indeed, Xu et al. showed that, in Chinese patients, macrophage inflammatory protein (MIP)-1β is an important chemokine in the intraocular environment of FU [20]. However, it should be noted that there is currently no universal gold standard for the diagnosis of FU, as evidenced by the diagnostic and classification criteria recently proposed by Caucasian and Asian authors [21,22].

The differential diagnosis between CMV and RV anterior uveitis is sometimes challenging; in these cases, performing an aqueous tap for the analysis of aqueous humor to search for CMV and RV antibody responses is essential to planning targeted treatments. In fact, this aqueous tap consistently affects therapeutic behavior, because the management of CMV anterior uveitis involves the use of topical or systemic antivirals, topical steroids, and topical antiglaucoma medications, while, in RV anterior uveitis, topical steroid drugs are not indicated, because inflammation is low-grade and because they may also speed up cataract and glaucoma formation. The treatment of RV anterior uveitis should therefore aim only to control IOP with antiglaucoma medications and/or laser or surgical procedures.

However, the literature seems to consider CMV and RV anterior uveitis as mutually exclusive, even if clinical manifestations of these uveitides can overlap: in other words, if an eye is affected by CMV, it cannot be considered affected by RV. In this study, we set out to demonstrate the possible coexistence of both CMV and RV in causing certain hypertensive VAU by analyzing the intraocular antibody production.

## 2. Materials and Methods

This retrospective study included consecutive patients with VAU of uncertain origin who were referred to the Ocular Immunology Unit, Azienda USL-IRCCS di Reggio Emilia, Italy, between January 2019 and March 2022.

All subjects had to meet the following criteria to be considered in the study:-recurrent unilateral granulomatous hypertensive anterior uveitis (IOP > 21 mmHg) without posterior synechiae or sectoral iris atrophy or epithelial–stromal keratitis;-negative QuantiFERON©-TB Gold and TPHA-VDRL tests;-normal serum lysozyme and angiotensin-converting enzyme levels;-execution of an anterior chamber paracentesis in the affected eye for laboratory tests during the active phase of uveitis; additionally, patients had not received topical or systemic antivirals or steroids for at least 2 weeks before the anterior chamber tap.

The laboratory tests on the extracted aqueous humor were the antibody index (AI) for RV as a parameter of the intraocular synthesis of specific antibodies against RV and the PCR or the AI for CMV.

Quantitative CMV DNA was amplified from aqueous humor by real-time PCR according to the manufacturer’s protocols (CMV ELITe MGB^®^ Kit).

Intraocular fluid and serum were analyzed with immunochemical nephelometry (Siemens, Germany) to quantify the albumin and total immunoglobulin G (IgG). Antigen-specific IgG against CMV and RV were measured using a commercially available one-point quantification ELISA assay (Enzygnost, Siemens, Germany) according to the manufacturer’s instructions. A specific AI is a modified Goldmann-Witmer coefficient (GWC) that was calculated for CMV and RV, as previously described [23]. Briefly, the measured optical density was evaluated as an arbitrary unit by reference to a standard curve. After multiplication with the dilution factor, we calculated the specific antibody quotient, QIgGspec = IgGspec (aqueous humor)/IgGspec (serum), and total antibody quotient QIgGtot = IgGtot (aqueous humor)/IgGtot (serum) using the aqueous humor and serum antibody concentrations. To determine the AI, it is necessary to calculate the Qlim (upper limit in the Reiber quotient diagram), which represents the discrimination line defined as 0 mg/L of the local synthesis of the antibodies, i.e., the maximum value of passively filtered immunoglobulins from the serum in certain conditions of the barrier state [24]. In general, the cases without the local synthesis of IgG in aqueous humor are below this line. Therefore, specific AI represent an evolution of the GWC, considering the permeability of the blood–ocular barrier. In our study, intraocular antibody production was considered positive if it was detected, regardless of the value.

The clinical features considered in the statistical analysis were the age at diagnosis in years, the diagnostic delay in months, sex, IOP, endothelial cell count, the presence of endotheliitis, iris atrophy, hypochromia, stellate precipitates, coin-shaped precipitates, Koeppe’s nodules, cataract, and anterior vitritis.

The study was conducted in agreement with the principles of the Declaration of Helsinki and received approval by the local ethics committee (protocol n. 0068784/2019 Comitato Etico dell’Area Vasta Emilia Nord, Italy).

### Statistical Analyses

The continuous variables are expressed as the mean ± standard deviation (SD) and the categorical variables as absolute frequencies and percentages. Comparisons among the groups were performed using Student’s *t*-tests and chi-square tests. A *p*-value ≤ 0.05 was considered significant. Statistical analysis was carried out using SPSS v. 28 (IBM Statistics).

## 3. Results

Forty-three eyes of as many subjects with VAU of uncertain origin were included. All patients (27 males (62.8%) and 16 females (37.2%) with a mean age at diagnosis of 53 ± 15 years) were Caucasian. The aqueous humor analysis revealed the presence of CMV in each case, with the following differences: 32 patients (74.4%) had a PCR or antibody response to CMV only, while 11 cases (25.6%) had an antibody response to both CMV and RV. The demographic, laboratory, and clinical features of the two groups are summarized in Table 1. These two groups were comparable in terms of age at diagnosis, diagnostic delay, and sex. The pure CMV group had 12/32 (37.5%) positive PCR and 29/32 (90.6%) positive AI, while the overlapping CMV-RV group had 2/11 (18.1%) positive CMV-PCR, 11/11 (100%) positive CMV-AI, and 11/11 (100%) positive RV-AI. The values for AI and the corresponding GWC are listed in Appendix A (Table A1 for the pure CMV group and Table A2 for the overlapping CMV-RV group). There were no statistically significant differences between the two groups in terms of the IOP, endothelial cell count, presence of endotheliitis, iris atrophy, stellate precipitates, coin-shaped precipitates, Koeppe’s nodules, or cataract. The only differences were hypochromia, which was much more present in the overlapping CMV-RV group (*p*-value: 0.02), and above all, the presence of anterior vitritis, highly predictive of the same group (*p*-value < 0.001).

## 4. Discussion

In this retrospective study, we analyzed the laboratory and clinical data of 43 patients with VAU of uncertain origin, demonstrating the possibility of the simultaneous presence of intraocular antibodies against CMV and RV and identifying some clinical findings: hypochromia and anterior vitritis, which allow a distinction from pure CMV uveitis.

CMV has a spectrum of ocular signs. However, along with usually being unilateral, it has been described that the presence of a few coin-shaped KPs has a positive predictive value of 90.9% for CMV [25]. In every CMV anterior uveitis, the pupil remains round, and posterior synechiae are absent. Vitreous inflammation is mild or typically absent. In some cases, the uveitis may be complicated by corneal endotheliitis, in which the endothelial cells are the primary target of CMV infection. Immune ring formation may be seen in CMV endotheliitis [18].

Interestingly, the literature shows differences between Asian and European patients in terms of clinical presentation. Chronic CMV anterior uveitis in the eyes of Asian patients resembles FU, while European patients have fewer KPs, which are located inferiorly and are characteristically coin-shaped. Therefore, CMV anterior uveitis is one of the most challenging diagnoses in immunocompetent patients; we previously reported that laboratory analyses using PCR and AI are useful and complementary to improving the diagnostic accuracy for CMV anterior uveitis [26]. In our report, 37.5% and 90.6% of patients belonging to the pure CMV group had positive PCR and positive AI for CMV, respectively. These results highlight the greater sensitivity of antibody detection in aqueous humor compared to PCR and confirm those reported in previous studies [26,27].

IOP increases during the course of the disease, with the maximum IOP in this type of uveitis generally higher than that in HSV or VZV anterior uveitis. Secondary glaucoma requiring surgery is the most common complication, followed by posterior subcapsular cataract due to chronic uveitis or to the use of topical corticosteroids to reduce intraocular inflammation [18].

FU is characterized by chronic low-grade inflammation involving the anterior uvea and vitreous humor [13]. The diagnosis of FU is clinical, which explains why most of these uveitides do not need the support of an aqueous humor analysis; however, it has been widely demonstrated that RV is the main causative agent of FU, especially in the Caucasian population [13,15,28,29,30]. One of the most peculiar findings is the presence of white small-to-medium-sized stellate KPs that are characteristically distributed diffusely over the endothelium [31]. The major iris finding in FU patients is heterochromia, generally subtle or absent in dark or brown irises, whereas it is prominent in light-colored ones, even if heterochromia may be present in less than 40% of patients with FU [32]. Iris nodules have been observed on the iris surface (Busacca nodules) or at the pupillary margin (Koeppe’s nodules) in about 20–30% of FU in a cases series [32,33,34]. Another iris feature is the absence of posterior synechiae [35]. Although FU is most commonly unilateral, bilateral involvement is possible (10%), detected almost always at the baseline rather than during a follow-up [32]. Since low-to-moderate vitritis is seen in the vast majority of FU patients, it should be considered a major diagnostic element [29,36]. Gradual progression of the disease is associated with cataract formation and glaucoma. Cataract is the most common complication, usually presenting in its posterior subcapsular form. Hence, FU diagnosis should be excluded in any young patient with unilateral lens opacification and no history of trauma or steroid use [37]. The elevations in IOP are initially intermittent but can later become chronic, and secondary glaucoma has been frequently reported in FU patients [38].

CMV and RV both represent an important cause of anterior uveitis and present common features, such as the absence of posterior synechiae and the presence of subcapsular cataract, also due to the inappropriate use of topical steroids. Clinical phenotypes can vary widely among both viruses, but it is not uncommon in clinical practice to find features simultaneously predictive of both CMV and RV anterior uveitis. Specifically, it is possible to find areas of iris atrophy that determine a more or less evident hypochromia, the presence of subcapsular cataract, or a few coin-shaped KPs mixed with stellate KPs (Figure 1). Our study found that anterior vitritis was almost completely absent in the pure CMV group, while it was present in the overlapping CMV-RV one (Figure 2); this could be a sign of the presence of RV, which is the main virus to determine the anterior vitritis among the viruses that induce VAU. These overlapping uveitides have never been described as a separate entity but have always traced back to a specific type of uveitis. In our case series, AI demonstrates the simultaneous presence of CMV and RV. We speculate that both viruses contribute to determining these particular forms of anterior uveitis. The practical implication is that uveitis that would not have been treated if they had been RV anterior uveitis alone would nevertheless benefit from the treatment against CMV; this clinical improvement would otherwise have not been achieved. Our results reiterate the importance of performing not only molecular diagnostics with PCR but also of searching for the antibody response—in particular, in the AI form—to increase the diagnostic sensitivity [16,26,27,28].

To the best of our knowledge, this is the first study that has highlighted the possible coexistence of both CMV and RV in causing overlapping VAU. However, this study has certain limitations. First, this was a retrospective analysis, which limited the consistency of the data. Second, the small sample size decreased the power of our statistical analysis. Third, this was a single-center study. Fourth, the two groups were not homogeneous—in particular, a small number of the overlapping CMV-RV group compared to the pure CMV one. Further prospective multicenter studies considering randomized groups and also including pure RV anterior uveitis are needed to confirm our results.

## 5. Conclusions

VAU could derive from simultaneous intraocular immune responses against CMV and RV, demonstrating a possible “converged” immune pathway following a variety of stimuli.

## Figures and Tables

**Figure 1 medicina-58-01054-f001:**
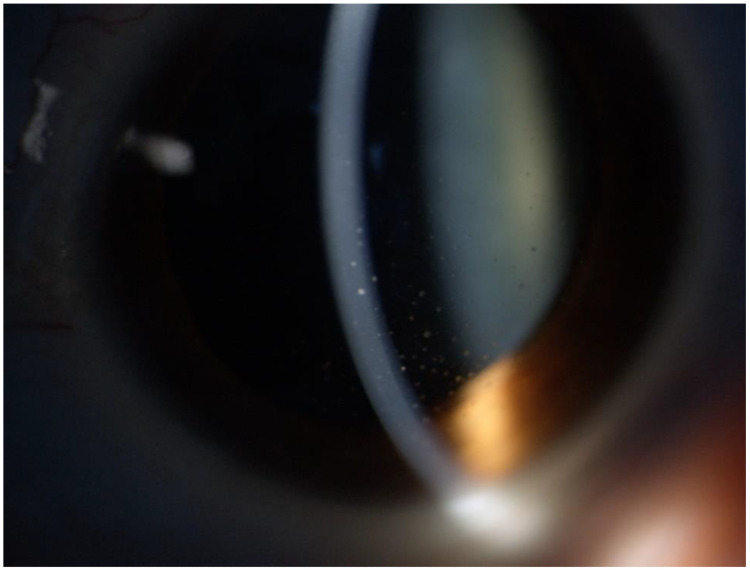
Few coin-shaped KPs mixed with stellate KPs in the overlapping CMV-RV group.

**Figure 2 medicina-58-01054-f002:**
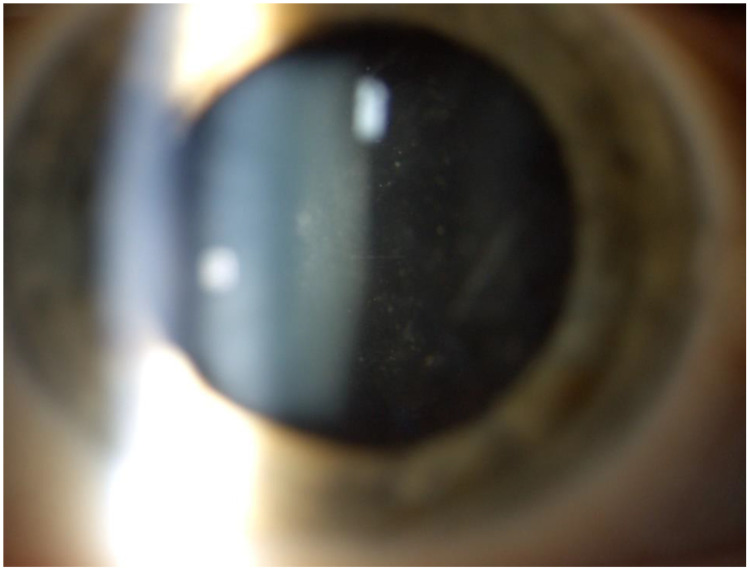
Anterior vitritis in the overlapping CMV-RV group.

**Table 1 medicina-58-01054-t001:** Descriptive table of demographic, laboratory, and clinical characteristics. Data are presented as the mean values ± SD or *n* (%); ns: not statistically significant; CMV: cytomegalovirus; RV: rubella virus; PCR: polymerase chain reaction; AI: antibody index; IOP: intraocular pressure.

		Pure CMV Anterior Uveitis (*n* = 32)	Overlapping CMV-RV Anterior Uveitis (*n* = 11)	
		Mean ± SD or *n* (%)	Mean ± SD or *n* (%)	*p*-Value
**Age at Diagnosis, Years**		56 ± 15	50 ± 15	ns
Diagnostic delay, months		126 ± 154	67 ± 66	ns
Sex	male	19 (59.4)	8 (72.7)	ns
	female	13 (40.6)	3 (27.3)	ns
CMV-PCR (+)		12 (37.5)	2 (18.2)	ns
CMV-AI (+)		29 (90.6)	11 (100)	ns
CMV-AI value		12.8 ± 22	10.0 ± 15.2	ns
IOP, mmHg		30.8 ± 6.9	27 ± 3.2	ns
Endothelial cell count, cells/mm^2^		2204 ± 578	2016 ± 742	ns
Endotheliitis		10 (31.3)	3 (27.3)	ns
Iris atrophy		2 (6.3)	3 (27.3)	0.06
Hypochromia		4 (12.5)	5 (45.5)	0.02
Stellate precipitate		0 (0)	1 (11.1)	ns
Coin-shaped precipitate		13 (48.1)	4 (44.4)	ns
Koeppe’s nodules		0 (0)	1 (9.1)	ns
Cataract		17 (53.1)	8 (72.7)	ns
Anterior vitritis		2 (6.3)	6 (54.5)	<0.001

## Data Availability

The data are contained within the article or in Appendix A.

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
