# Peer review of "Aqueous Humor Analysis in Overlapping Clinical Diagnosis of Cytomegalovirus and Rubella Virus Anterior Uveitis"

_medicina, 2022, doi:10.3390/medicina58081054_

Round 1

Reviewer 1 Report

The authors performed a retrospective study to investigate the etiology and describe the clinical features of certain viral anterior uveitis. However, there are something fundamentally wrong

1. Authors confused rubella virus (RV)-associated uveitis with Fuchs syndrome. They deemed those subjects with positive antibody response to both cytomegalovirus (CMV) and RV as the CMV-Fuchs group. The diagnosis of Fuchs syndrome is a clinical one. It is not necessary to do aqueous humor analysis for diagnostic purpose. Although it is reported that rubella virus could be involved in the pathogenesis of Fuchs syndrome, the infection of rubella virus does not mean the diagnosis of Fuchs syndrome.

2. I can t see the ethical statement of this study. This should be clarified in the manuscript.

3. This study lacks some essential and fundamental information of collected patients. Are these patients Caucasian? Did these patients received systemic antivirals or topical steroids treatments? Did authors collect samples from active patients?

4. The authors stated that this study collected patients with hypertensive viral anterior uveitis. However, the intraocular pressure of subjects is not mentioned in the manuscript.

5. Authors only detected specific antibodies against CMV and RV without measuring other antivirus antibodies, such as herpes simplex virus or varicella-zoster virus. Why didnt authors perform theses antibodies detection?

6. In the Results section, authors should list the exact value of detected antibody concentrations and the calculated Goldmann-Witmer coefficient.

Author Response

The authors performed a retrospective study to investigate the etiology and describe the clinical features of certain viral anterior uveitis. However, there are something fundamentally wrong. 

Point 1. Authors confused rubella virus (RV)-associated uveitis with Fuchs’ syndrome. They deemed those subjects with positive antibody response to both cytomegalovirus (CMV) and RV as the CMV-Fuchs group. The diagnosis of Fuchs’ syndrome is a clinical one. It is not necessary to do aqueous humor analysis for diagnostic purpose. Although it is reported that rubella virus could be involved in the pathogenesis of Fuchs’ syndrome, the infection of rubella virus does not mean the diagnosis of Fuchs’ syndrome.

Response 1. We thank the reviewer for pointing this out. Although we believe that rubella virus (RV) is the main culprit of Fuchs uveitis, especially in the Caucasian population, we have changed Fuchs uveitis to RV anterior uveitis and we have consequently changed the title as well.

Point 2. I can’t see the ethical statement of this study. This should be clarified in the manuscript.

Response 2. Ethical statement has been placed in the dedicated part at the end of the manuscript as indicated in the template (see “Institutional Review Board Statement”); in any case we have added the sentence also inside the manuscript, in the materials and methods section

Point 3. This study lacks some essential and fundamental information of collected patients. Are these patients Caucasian? Did these patients received systemic antivirals or topical steroids treatments? Did authors collect samples from active patients?

Response 3. All patients included in the study are Caucasian, they had not received topical antivirals or steroids for at least 2 weeks before anterior chamber tap and we collect samples from patients in the acute phase of the uveitis.

Point 4. The authors stated that this study collected patients with hypertensive viral anterior uveitis. However, the intraocular pressure of subjects is not mentioned in the manuscript.

Response 4. We appreciate the reviewer for highlighting these missing data. We have added the inclusion criteria “intraocular pressure (IOP) > 22 mmHg” at the time of aqueous humor tap in the section “Materials and methods” and the IOP mean and standard deviation values in Table 1.

Point 5. Authors only detected specific antibodies against CMV and RV without measuring other antivirus antibodies, such as herpes simplex virus or varicella-zoster virus. Why didn’t authors perform theses antibodies detection?

Response 5. The reviewer rightly underlined that we researched only specific antibodies against CMV and RV without measuring other antivirus antibodies, such as herpes simplex virus or varicella-zoster virus. We searched only for these antibodies because we based on the clinical picture of hypertensive anterior uveitis which had characteristics of both RV and CMV, but not of HSV or VZV, specifically there were no synechiae, nor typical sectoral iris atrophy, nor stromal keratitis. Secondly, the volume collected during a single aqueous tap does not normally allow to calculate the antibody index for all viruses, but only for some targeted ones. Therefore, we should have performed more than one tap, but we did not consider it to be ethically correct since we had no clinical suspicion

Point 6. In the Results section, authors should list the exact value of detected antibody concentrations and the calculated Goldmann-Witmer coefficient.

Response 6. We have added a table with the values of the detected antibody concentrations and the calculated Goldmann-Witmer coefficient in the Appendix as Supplementary data.

Reviewer 2 Report

Review Report:

General Comments:

This is a retrospective single-center study investigating the etiology of viral anterior uveitis (VAU) in the hopes of improving the diagnostic approach in cases where the clinical features are not distinctive of cytomegalovirus (CMV) versus rubella virus (RV). To do this, the authors reviewed results from the anterior chamber paracentesis of patients with VAU of uncertain origin at the ocular immunology clinic in which mainly CMV PCR and antibody index for CMV and RV were done. In addition, they investigated differences in ocular findings between the groups with differing etiologies (pure CMV vs CMV/RV VAU). The major findings of this study were: (1) co-presence of CMV and RV in 11/43 of the cases of VAU of uncertain origin whereas the other 32/43 eyes had a pure CMV VAU and (2) the CMV-RV overlapping group had a statistically significant higher rate of hypochromic and anterior uveitis.

Diagnosis and identifying the etiology of VAU would help guide and improve management. This study highlights the prevalence of co-existing etiologies of both CMV and RV and identifies clinical findings that allow distinction from purely CMV VAU. This would highlight that these etiologies are not mutually exclusive which should be taken into consideration when making a differential diagnosis. This would then allow for guided therapy to improve outcomes in these patients. Moreover, the paper is concise and overall well-written with regard to language.

Kindly find the following suggestions to review:

Title Section:

-          Please double-check the title in the main document as it appears to be incomplete and different from the title in the submission. However, this could be a formatting error.

-          Please adjust the superscript for the senior author to be numerically ascending (i.e. 1,6 instead of 6,1)

Introduction:

-          Paragraph 2, Line 10: Consider changing “…cause FU in Asian population…” to “cause FU in the Asian population…”

-          Paragraph 4, line 2: Consider changing “…of these uveitis...” to “… of these uveitides...”

Methods:

-          The paper would benefit from a more specific methodology with regards to inclusion and the criteria used. Please specify and indicate how were the patients with VAU of unspecified origin identified and selected.

-          Consider including the anterior chamber paracentesis procedure as part of the inclusion criteria. The way it is stated in the 2nd sentence of the methods makes it seem that the paracentesis was done during the study rather than being retrospectively reviewed.

-          Please specify which t-test was performed for the continuous variables.

Results:

-          While the results were concise, consider starting off with the overall demographics (age, race, sex) of the entire study population so that it can be used to relate to other studies on VAU.

-          Please include percentages next to the numbers of the reported sub-groups and proportions (i.e. pure CMV, overlapping CMV/RV, PCR/AI positivity)

-          Please be consistent with the terminology regarding the p-value. It is written as “p-value” when comparing hypocromia then as “p <” in the line below it.

Discussion:

-          Consider starting the discussion with a one-sentence “take-away” summary of the methods and results of the study and results as this would help the reader as they read the rest of the paper.

-          Please discuss the results with regard to PCR and AI positivity rates seen in this study for CMV and how it relates to reports in the literature. While this is not the main purpose of this study, it would add validity to show that the rates were in line with the literature.

-          Please add a reference to the statement in the 2nd paragraph which says: “IOP increases during the course of the disease, with maximum IOP in this type of uveitis generally higher than that in HSV or VZV anterior uveitis. Secondary glaucoma requiring surgery is the most common complication, followed by posterior subcapsular cataract due to chronic uveitis or to the use of topical corticosteroids to reduce intraocular inflammation.”

Figures and Tables:

-          Table 1: Please reduce the number of columns to 1 per group. The mean +/- SD and n (%) can be reported on the same column for each group without the need to separate them into 4. In addition, in the majority of variables with only 2 measures, there is no need to report both presence and absence as they are mutually exclusive. Reporting only the presence for these variables would help tidy up the table and make it easier to read.

Author Response

General Comments:

This is a retrospective single-center study investigating the etiology of viral anterior uveitis (VAU) in the hopes of improving the diagnostic approach in cases where the clinical features are not distinctive of cytomegalovirus (CMV) versus rubella virus (RV). To do this, the authors reviewed results from the anterior chamber paracentesis of patients with VAU of uncertain origin at the ocular immunology clinic in which mainly CMV PCR and antibody index for CMV and RV were done. In addition, they investigated differences in ocular findings between the groups with differing etiologies (pure CMV vs CMV/RV VAU). The major findings of this study were: (1) co-presence of CMV and RV in 11/43 of the cases of VAU of uncertain origin whereas the other 32/43 eyes had a pure CMV VAU and (2) the CMV-RV overlapping group had a statistically significant higher rate of hypocromia and anterior uveitis.

Diagnosis and identifying the etiology of VAU would help guide and improve management. This study highlights the prevalence of co-existing etiologies of both CMV and RV and identifies clinical findings that allow distinction from purely CMV VAU. This would highlight that these etiologies are not mutually exclusive which should be taken into consideration when making a differential diagnosis. This would then allow for guided therapy to improve outcomes in these patients. Moreover, the paper is concise and overall well-written with regard to language.

Kindly find the following suggestions to review:

Point 1. Title Section:

  • Please double-check the title in the main document as it appears to be incomplete and different from the title in the submission. However, this could be a formatting error.
  • Please adjust the superscript for the senior author to be numerically ascending (i.e. 1,6 instead of 6,1)

Response 1. We thank the reviewer for the valuable suggestions: the title was a formatting error and we adjust the superscript for the senior author to be numerically ascending.

Point 2. Introduction:

  • Paragraph 2, Line 10: Consider changing “…cause FU in Asian population…” to “cause FU in the Asian population…”
  • Paragraph 4, line 2: Consider changing “…of these uveitis...” to “… of these uveitides...”

Response 2. We have corrected the sentences reported as rightly indicated by the reviewer.

Point 3. Methods:

  • The paper would benefit from a more specific methodology with regards to inclusion and the criteria used. Please specify and indicate how were the patients with VAU of unspecified origin identified and selected.
  • Consider including the anterior chamber paracentesis procedure as part of the inclusion criteria. The way it is stated in the 2nd sentence of the methods makes it seem that the paracentesis was done during the study rather than being retrospectively reviewed.
  • Please specify which t-test was performed for the continuous variables.

Response 3. We have expanded and modified the section “Materials and methods” by adding inclusion criteria, according to reviewer suggestions. Furthermore, we specify that the t-test performed for the continuous variables was Student's t-test.

Point 4. Results:

  • While the results were concise, consider starting off with the overall demographics (age, race, sex) of the entire study population so that it can be used to relate to other studies on VAU.
  • Please include percentages next to the numbers of the reported sub-groups and proportions (i.e. pure CMV, overlapping CMV/RV, PCR/AI positivity)
  • Please be consistent with the terminology regarding the p-value. It is written as “p value” when comparing hypocromia then as “p <” in the line below it.

Response 4. We appreciate the reviewer for these suggestions. We have expanded the section “Results” and corrected the terminology regarding the p-value.

Point 5. Discussion:

  • Consider starting the discussion with a one-sentence “take-away” summary of the methods and results of the study and results as this would help the reader as they read the rest of the paper.
  • Please discuss the results with regard to PCR and AI positivity rates seen in this study for CMV and how it relates to reports in the literature. While this is not the main purpose of this study, it would add validity to show that the rates were in line with the literature.
  • Please add a reference to the statement in the 2nd paragraph which says: “IOP increases during the course of the disease, with maximum IOP in this type of uveitis generally higher than that in HSV or VZV anterior uveitis. Secondary glaucoma requiring surgery is the most common complication, followed by posterior subcapsular cataract due to chronic uveitis or to the use of topical corticosteroids to reduce intraocular inflammation.”

Response 5. We thank the reviewer for the valuable tips. We have corrected the section “Discussion” according to reviewer suggestions.

Point 6. Figures and Tables:

  • Table 1: Please reduce the number of columns to 1 per group. The mean +/- SD and n (%) can be reported on the same column for each group without the need to separate them into 4. In addition, in the majority of variables with only 2 measures, there is no need to report both presence and absence as they are mutually exclusive. Reporting only the presence for these variables would help tidy up the table and make it easier to read.

Response 6. We have corrected the table as indicated by the reviewer.

Round 2

Reviewer 1 Report

The authors have made a significant effort to answer my comments. However, there are several points which could be improved.

1. In the revised manuscript, the authors stated they collected samples from patients in the acute phase. The diagnosis of most uveitis entities depends on clinical characteristics. It is not ethical to collect aqueous samples from patients for diagnostic purpose,especially in acute phase. Authors should point out a reasonable and ethical purpose of this study for anterior chamber paracentesis.

2. When mentioning Fuchs’ syndrome in the Asian, it would improve the manuscript if authors provide more updated references (PMID: 34108225 and 31776091).

3. Authors have added a table with the values of the detected antibody concentrations and GWC. However, the format is inconsistent. Why did they list one value for some parents,while list two values for the others?

Author Response

The authors have made a significant effort to answer my comments. However, there are several points which could be improved.

1. In the revised manuscript, the authors stated they collected samples from patients in the acute phase. The diagnosis of most uveitis entities depends on clinical characteristics. It is not ethical to collect aqueous samples from patients for diagnostic purpose, especially in acute phase. Authors should point out a reasonable and ethical purpose of this study for anterior chamber paracentesis.

Response 1. We thank the reviewer for this important question. We collected aqueous humor samples from patients in the acute phase to be analyzed because aqueous tap consistently affects therapeutic behavior. As we discussed in the manuscript, management of CMV anterior uveitis involves the use of topical or systemic antivirals, topical steroids, and topical anti-glaucoma medications, while in RV anterior uveitis, topical steroid drugs are not indicated because inflammation is low-grade and because they also may speed up cataract and glaucoma formation. Therefore, since differential diagnosis between CMV and RV anterior uveitis is sometimes challenging, performing an aqueous tap to search for CMV and RV is essential to planning targeted treatment. In particular, overlapping CMV-RV uveitis would not have been treated if they had been considered RV anterior uveitis alone, so they would nevertheless benefit from the treatment against CMV; this clinical improvement would otherwise not be achieved. We think aqueous tap in the acute phase is a reasonable and ethical choice and it shouldn't be limited to being obtained during cataract surgery for research studies.

2. When mentioning Fuchs’ syndrome in the Asian, it would improve the manuscript if authors provide more updated references (PMID: 34108225 and 31776091).

Response 2. We appreciate the reviewer for these valuable references that enhance our manuscript. We have included these references in the introduction to improve it.

3. Authors have added a table with the values of the detected antibody concentrations and GWC. However, the format is inconsistent. Why did they list one value for some parents, while list two values for the others?

Response 3. We thank the reviewer for this question. We distinguished Antibody Index (AI) from Goldmann-Witmer coefficient (GWC) because they are two different indices which sometimes can have the same numerical value, while other times they differ; therefore, we list one value for some patients, while we list two values for the others. In particular, GWC specific for a pathogen is expressed by the ratio between the ratio of the concentrations of specific antibodies in aqueous humor and serum (QIgG spec) and the ratio of the concentrations of total IgG in the aqueous humor and serum (QIgG tot):

GWC = QIgG spec / QIgG tot

The expression of the IgG concentration ratio has a limit in the event that there is moderate or severe damage to the blood-ocular barrier. In these cases, GWC can give false positives. This can be overcome by using specific AI which represents an evolution of the GWC considering the permeability of the blood-ocular barrier. The barrier damage is a function of the ratio of albumin concentration in aqueous humor and in serum:

Qalb = Albaqueous / Albserum.

For the determination of the AI it is necessary to calculate the Qlim, given by the equation:

Qlim = [0.93 x (Qalb2 + 6 x 10-6)0,5 - 1.7] x 10-3

Indeed, when QIgG tot < Qlim, AI is calculated as QIgG spec/ QIgG tot. When QIgG tot > Qlim, the corrected AI is calculated with AI = QIgG spec/Qlim

For all these reasons, AI and GWC values might be different.
